# Beyond Noisy-TVs: Noise-Robust Exploration Via Learning Progress Monitoring

**Zhibo Hou, Zhiyu An, Wan Du**
Department of Computer Science and Engineering
University of California, Merced
`{zhou6,zan7,wdu3}@ucmerced.edu`

## Abstract

When there exists an unlearnable source of randomness (noisy-TV) in the environment, a naively intrinsic reward driven exploring agent gets stuck at that source of randomness and fails at exploration. Intrinsic reward based on uncertainty estimation or distribution similarity, while eventually escapes noisy-TVs as time unfolds, suffers from poor sample efficiency and high computational cost. Inspired by recent findings from neuroscience that humans monitor their improvements during exploration, we propose a novel method for intrinsically-motivated exploration, named Learning Progress Monitoring (LPM). During exploration, LPM rewards model improvements instead of prediction error or novelty, effectively rewards the agent for observing learnable transitions rather than the unlearnable transitions. We introduce a dual-network design that uses an error model to predict the expected prediction error of the dynamics model in its previous iteration, and use the difference between the model errors of the current iteration and previous iteration to guide exploration. We theoretically show that the intrinsic reward of LPM is zero-equivariant and a monotone indicator of Information Gain (IG), and that the error model is necessary to achieve monotonicity correspondence with IG. We empirically compared LPM against state-of-the-art baselines in noisy environments based on MNIST, 3D maze with 160x120 RGB inputs, and Atari. Results show that LPM's intrinsic reward converges faster, explores more states in the maze experiment, and achieves higher extrinsic reward in Atari. This conceptually simple approach marks a shift-of-paradigm of noise-robust exploration.

## 1 Introduction

A major obstacle to efficient exploration in complex environment is the sparse reward problem (Randløv & Alstrøm, 1998; Pathak et al., 2017). In many real world tasks, extrinsic reward are rare and delayed, for instance agents may only receive a positive reward signal only after reaching the exit of a maze, or a robot arm requires completing a long sequence of actions before a reward is given. In such settings, random exploration become highly inefficient because the agent receives little to no guidance from the environment. To address this challenge, researchers have introduced intrinsic reward (Schmidhuber, 1991; Thrun, 1992; Ladosz et al., 2022), which act as additional internal reward signals during exploration. Curiosity-driven exploration methods (Burda et al., 2018; Pathak et al., 2017) use prediction error or uncertainty estimation to generate intrinsic rewards, while episodic bonus methods (Badia et al., 2020; Henaff et al., 2022; Zhang et al., 2021) encourage exploration by rewarding state novelty. These intrinsic reward based methods encourage the agent to explore systematically even in absence of extrinsic feedback, thereby improving exploration efficiency and enabling progress on sparse reward tasks.

While promising, this class of methods suffers from a long-standing challenge known as the noisy-TV problem (Burda et al., 2018). For example, a naively curious agent may become fixated on the unpredictability of static noise on a TV screen, wasting time gathering observations that do not improve its world model. Despite being a well-known issue (Schmidhuber, 1991; Burda et al., 2019), robust exploration in the presence of noisy-TVs remains an open problem. Existing works (Mavor-Parker et al., 2022; Wang et al., 2024; Jiang et al., 2025) focused on isolating the uncertainty that can

---

Code available at `https://github.com/Akuna23Matata/LPM_exploration`

be reduced by more data (known as epistemic uncertainty) from the uncertainty that is irreducible by more data (aleatoric uncertainty), and filter the intrinsic reward signals to only encourage exploring the former type of uncertainty, which is also referred as novelty. However, distinguishing these two types of uncertainty is extremely challenging. Most approaches require either a strong prior over the state space (Wang et al., 2024; Jiang et al., 2025; Henaff et al., 2023; Zhang et al., 2021) or massive amounts of data (Burda et al., 2018; Mavor-Parker et al., 2022) before the agent can reliably filter out noise. This slow convergence causes early exploration to be dominated by noise, wasting large numbers of samples and limiting the agent's ability to explore effectively in complex, real-world environments.

Recent neuroscience findings (Ten et al., 2021) demonstrate that humans naturally monitor *learning progress* during exploration. During exploration, humans tend to observe the transitions that makes them learn the most about the dynamics. Because observing an unlearnable transition does not produce learning progress, this strategy is naturally robust to noisy-TV distractions and promotes efficient exploration by directing effort toward the most informative experiences. In the context of intrinsic reward, this insight suggests that we should reward model improvement rather than prediction error or state novelty. Inspired by this study, we propose a fundamentally different approach to intrinsic motivation, called **Learning Progress Monitoring (LPM)**. In this work, we first develop a formal definition of the noisy-TV problem, then introduce our solution and complete algorithm, LPM, as a new form of intrinsic reward. We then provide a theoretical analysis showing that our approach is monotonically related to information gain. Finally, we conduct comprehensive experiments to evaluate the empirical performance of LPM and demonstrate its advantages over existing intrinsic motivation methods. We summarize our contributions as the followings:

- We propose a novel intrinsic motivation-driven exploration method, Learning Progress Monitoring, inspired by recent neuroscience research findings.
- We provide a theoretical analysis showing that LPM is zero-equivariant and a monotone indicator of information gain, and that our dual-network design is necessary to achieve such properties.
- We conduct comprehensive experiments and demonstrate the superior efficiency and noise robustness of our method compared with the state-of-the-art baselines.

## 2 BACKGROUND

### 2.1 REINFORCEMENT LEARNING AND MATHEMATICAL FORMULATION.

We consider the reinforcement learning (RL) problem formalized as a *Partially Observable Markov Decision Process (POMDP)* defined by the tuple $\mathcal{M} = (\mathcal{S}, \mathcal{A}, \mathcal{T}, r, \Omega, O, \gamma)$, where $\mathcal{S}$ is the state space, $\mathcal{A}$ is the action space, $\mathcal{T} : \mathcal{S} \times \mathcal{A} \times \mathcal{S} \to [0,1]$ is the transition probability function, $r : \mathcal{S} \times \mathcal{A} \to \mathbb{R}$ is the reward function, $\Omega$ is the observation space, $O : \mathcal{S} \times \mathcal{A} \times \Omega \to [0,1]$ is the observation function, where $O(o|s', a)$ gives the probability of receiving observation $o$ after taking action $a$ and transitioning to a new state $s'$, and $\gamma \in [0,1]$ is the discount factor.

The reinforcement learning agent's behavior is defined by a policy $\pi : \Omega \times \mathcal{A} \to [0,1]$, where $\pi(a|o)$ represents the probability of taking action $a$ given the current observation $o \in \Omega$. The objective in RL is to find an optimal policy $\pi^*$ that maximizes the expected cumulative discounted reward:

$$\pi^* = \arg\max_{\pi} \mathbb{E}_{\pi} \left[ \sum_{t=0}^{T} \gamma^t r(s_t, a_t) \right].$$

### 2.2 INTRINSIC MOTIVATION

Extrinsic reward is the environmental reward usually provided when agent accomplished certain task or achieved certain goal. In environments with sparse and delayed extrinsic rewards, exploration is highly inefficient because the agent receives little to no feedback about which behaviors lead to success. To address this challenge, intrinsic reward signal are introduced to motivate systematic exploration (Ryan & Deci, 2000; Burda et al., 2019; Badia et al., 2020). Formally, the agent's total reward at time step $t$ is given by:

$$r_t = r_t^{\text{e}} + \beta\, r_t^{\text{i}},$$

where $r_t^{\text{e}}$ is the extrinsic reward provided by the environment, $r_t^{\text{i}}$ is the intrinsic reward generated by the agent, and $\beta$ is a weighting coefficient that balances the two signals.

Intrinsic motivation methods differ primarily in how $r_t^i$ is defined. Curiosity-driven methods (Burda et al., 2018; Pathak et al., 2017) generate intrinsic rewards based on *prediction error* or *uncertainty estimation*, encouraging the agent to visit transitions that are surprising or poorly understood. In contrast, episodic bonus methods (Badia et al., 2020; Henaff et al., 2022; Zhang et al., 2021) focus on *state novelty*, rewarding the agent for visiting states that have not been encountered within a given episode. These intrinsic reward signals guide the agent to explore more effectively, enabling progress in sparse-reward tasks by providing continuous internal feedback.

### 2.3 THE NOISY TV PROBLEM

Intrinsic motivation may fail in environments with high stochasticity due to their inability to distinguish meaningful uncertainty or novelty from random noise. To formalize this, we distinguish between two types of uncertainty in the agent's predictive model of the environment dynamics. *Epistemic uncertainty* (also known as model uncertainty) arises from a lack of knowledge about the true underlying dynamics or observation model. It reflects the model's ignorance due to limited data and can be reduced by collecting more informative experience. *Aleatoric uncertainty* is inherent to the environment and arises from stochasticity in the dynamics or observation processes themselves. It reflects noise or randomness that cannot be reduced even with infinite data, e.g. sensor noise.

The *noisy-TV problem* (Burda et al., 2018) is a classic failure case when the intrinsic motivation mechanism confuses aleatoric uncertainty with epistemic uncertainty. The TV may already be on in the environment (*state noise*), or the agent may hold a remote and accidentally turn it on by pressing a button (*action-triggered noise*). In both cases, the resulting observations are highly unpredictable and contain no learnable structure. Formally, in a POMDP setting, the next observation, $o' = g(s', a) + \epsilon$, can be decomposed into a deterministic transition from next state and action, $g(s', a)$, and a unlearnable noise independent of environment dynamics, $\epsilon$. As the magnitude of $\epsilon$ increases, the observations become dominated by environment-independent randomness. Consequently, intrinsic motivation methods that rely on novelty or prediction error become increasingly driven by aleatoric uncertainty rather than epistemic uncertainty, causing the agent to focus on noise rather than meaningful transitions.

## 3 LEARNING PROGRESS MONITORING

We use $t$ to denote the environment time steps and $\tau$ to denote model updating steps, such that the $i$-th model updating step $\tau_i$ happens in the $(N \cdot i)$-th environment step $t_{N \cdot i}$, where $N$ is an integer that sets the length of model updating cycle. Then, we use $f_\theta^{(\tau)}$ to represent a model parameterized by $\theta$ after $\tau$-th model updating step.

Subsequently, we consider a model-based reinforcement learning setting, where the agent periodically updates a dynamics model $f_\theta$ that gives the prediction of next observation $\hat{o}_{t+1}$ given the current observation $o_t$ and action $a_t$: $\hat{o}_{t+1} = f_\theta^{(\tau)}(o_t, a_t)$. Let $\mathcal{B}$ consists of $(o_t, a_t, o_{t+1})$ be the replay buffer of transition dynamics used to fit $f_\theta$. Upon observing the true next observation $o_{t+1}$, we can compute the difference between the true and predicted next observations. We define the log Mean Squared Error (MSE) of $\tau$-th dynamics model's prediction at time step $t$ to be $\varepsilon_t^{(\tau)}$:

$$\varepsilon_t^{(\tau)}(o_{t+1}) = \log\left(\text{MSE}(o_{t+1}, \hat{o}_{t+1})\right) = \log\left(\frac{1}{\dim(\Omega)}\left\|o_{t+1} - f_\theta^{(\tau)}(o_t, a_t)\right\|_F^2\right) \tag{1}$$

Intuitively, $\varepsilon_t^{(\tau-1)}(o_{t+1}) - \varepsilon_t^{(\tau)}(o_{t+1})$ captures the prediction accuracy improvement that model $f_\theta$ gained from the $\tau$-th updating step. A naive method to estimate the said improvement would be to save $f_\theta^{(\tau-1)}$ and infer it on the current $(o_t, a_t)$ pair. Instead, we propose measuring the *expected error* of the last dynamics model $\mathbb{E}_\mathcal{D}[\varepsilon_t^{(\tau-1)}(o_{t+1})]$ and compare it with the error of the current model $\varepsilon_t^{(\tau)}(o_{t+1})$. In Section 4, we will show that this modification is the key to ensure the quality of the intrinsic reward signal. To measure the expected error, we introduce a fixed-size replay queue $\mathcal{D}$ with size $d$ where each entry consists of $(o_t, a_t, \varepsilon_t^{(\tau)}(o_{t+1}))$. Then, we introduce a separate *error model* $g_\phi \colon \mathcal{O} \times \mathcal{A} \to \mathbb{R}$, parameterized by $\phi$, which predicts the error of the dynamics model before the last updating step:

$$g_\phi^{(\tau)}(o_t, a_t) \approx \mathbb{E}_\mathcal{D}\left[\varepsilon_t^{(\tau-1)}(o_{t+1})\right] \tag{2}$$

The dynamics model $f_\theta$ and error model $g_\phi$ are updated simultaneously with the most up-to-date replay buffers $\mathcal{B}$ and $\mathcal{D}$, respectively. At each updating step $\tau$, because $\mathcal{D}$ only contains the prediction errors using the dynamics model that is updated up to the last updating step, $g_\phi^{(\tau)}$ is fit using the prediction errors of the last dynamics model $f_\theta^{(\tau-1)}$.

Building on the above design, we introduce our intrinsic reward formulation based on *learning progress monitoring* with method summarized in Algorithm 1:

$$r_t^i = \mathbb{E}_\mathcal{D}\left[\varepsilon_t^{(\tau-1)}(o_{t+1})\right] - \varepsilon_t^{(\tau)}(o_{t+1}) = g_\phi^{(\tau)}(o_t, a_t) - \varepsilon_t^{(\tau)}(o_{t+1}) \tag{3}$$

---

**Algorithm 1** Learning Progress Monitoring Exploration

---

**Require:** policy model $\pi$, dynamics model $f_\theta$, error model $g_\phi$, replay queue $\mathcal{D}$ with fixed size $d$, replay buffer $\mathcal{B}$, intrinsic reward weight $\beta$, update cycle $N$

1: **while** not converged **do**
2:     Observe $o_t$, sample $a_t \sim \pi(\cdot|o_t)$, execute $a_t$ in the environment and observe $o_{t+1}, r_t^e$
3:     Compute $\varepsilon_t^{(\tau)}(o_{t+1})$ by Equation 1
4:     Push $(o_t, a_t, o_{t+1})$ to replay buffer $\mathcal{B}$
5:     Push $(o_t, a_t, \varepsilon_t^{(\tau)}(o_{t+1}))$ to fixed-size queue $\mathcal{D}$
6:     $r_t^i = g_\phi^{(\tau)}(o_t, a_t) - \varepsilon_t^{(\tau)}(o_{t+1})$ if $|\mathcal{D}| = d$, else 0
7:     $r_t = r_t^e + \beta \cdot r_t^i$
8:     Update policy $\pi$ using $r_t$ and any RL algorithm
9:     **if** $t \bmod N = 0$ **then**
10:         Update $\tau$; Update $f_\theta \to \mathbb{E}_{\mathcal{B}(<\tau)}[o_{t+1}]$, $g_\phi \to \mathbb{E}_{\mathcal{D}(<\tau)}[\varepsilon_t^{(\tau-1)}(o_{t+1})]$
11:     **end if**
12: **end while**

---

## 4   Theoretical Analysis of LPM

To formally demonstrate the rationale behind our Learning Progress Monitoring (LPM) intrinsic reward, we provide the following two analyses:

1. The monotonicity and zero-equivariance relationships between $r^i$ and the information gain (IG) of the dynamics model parameters,
2. The necessity of using $g_\phi$ to measure the expected prediction error of the previous model.

### 4.1   Relationship between $r^i$ and Information Gain

We begin by defining information gain in a way consistent with our setup.

**Definition 1** (Information Gain). *Let $\theta \in \Theta$ be model parameters and $D$ a dataset. Let $p(\theta \mid D)$ denote the posterior distribution of $\theta$ given $D$, and let $p(D)$ denote the marginal likelihood $p(D) = \int_\Theta p(D \mid \theta)\, p(\theta)\, d\theta$. The information gain (IG) provided by the data $D$ about $\theta$ is defined as*

$$\text{IG} := \mathbb{E}_{p(\theta|D)}[\log p(D \mid \theta)] - \log p(D) = \text{KL}\big(p(\theta \mid D) \,\|\, p(\theta)\big).$$

*This measures the expected reduction in uncertainty about $\theta$ after observing $D$, and depends only on the marginal likelihood and posterior.*

Here, $p(D \mid \theta)$ denotes the likelihood of the dataset $D$ under model parameters $\theta$. Previous work also use KL–divergence as a metric for information gain (Bottero et al., 2022), which is equivalent to our definition. We assume an i.i.d. Gaussian observation model: $o_{t+1} = f_\theta(o_t, a_t) + \varepsilon_t, \varepsilon_t \sim \mathcal{N}(0, \sigma^2 I)$. Hence, for a dataset $D = \{(o_t, a_t, o_{t+1})\}_{t=1}^T$, $\log p(D \mid \theta) = -\frac{Td}{2}\log(2\pi\sigma^2) - \frac{Td}{2\sigma^2}\text{MSE}(\theta)$ where $d = \dim(\Omega)$ and $\text{MSE}(\theta) = \frac{1}{Td}\sum_{t=1}^T \|o_{t+1} - f_\theta(o_t, a_t)\|^2$. Note that the first term in the above $\log p(D \mid \theta)$ is a constant. For the rest of the analysis we will use the log–MSE surrogate $\log p(D \mid \theta) \approx -c \log \text{MSE}(\theta) + \text{const}(D)$, which preserves monotonicity with $\text{MSE}(\theta)$ while improving numerical stability of the reward.

Using this notion of IG, we can formalize the connection between LPM's intrinsic reward and the expected reduction in model uncertainty:

---

Derivation is in Appendix A.1.

**Theorem 4.1** (Monotonicity and Zero Equivalence). *Let $\theta \in \Theta$ be model parameters with prior $p(\theta)$ and posterior $p(\theta \mid D)$ given dataset $D$. Assume the likelihood depends on $\theta$ only through a positive scalar function $\mathrm{MSE}(\theta)$ such that $\log p(D \mid \theta) = -c \log \mathrm{MSE}(\theta) + \mathrm{const}(D), c > 0$. For intrinsic reward $r^i := \mathbb{E}_{p(\theta)}[\log \mathrm{MSE}(\theta)] - \log \mathrm{MSE}(\theta_D)$ where $\theta_D$ is a chosen point in $\Theta$ satisfying $\log p(D \mid \theta_D) \geq \mathbb{E}_{p(\theta|D)}[\log p(D \mid \theta)]$, Then the following hold:*

1. *$r_i \geq \frac{1}{c} \mathrm{IG}$, where IG is defined by Definition 1.*
2. *$\mathrm{IG} = 0 \implies r_i = 0$.*
3. *$r_i = 0 \implies \mathrm{IG} = 0$ under the identifiability condition that the likelihood is non-constant and injective in $\mathrm{MSE}(\theta)$.*

Proof is in Appendix A.3. In practice we set $\theta_D$ to the MLE (or an approximate likelihood maximizer); Lemma A.1 in the appendix shows this choice satisfies the required inequality. Theorem 4.1 establishes that the LPM intrinsic reward is a monotone indicator of the information gain of the dynamics model. Intuitively, a larger positive intrinsic reward corresponds to a larger expected reduction in uncertainty, and zero intrinsic reward occurs if and only if the model has learned nothing new, i.e., the information gain is zero.

## 4.2 NECESSITY OF ERROR MODEL $g_\phi$

The second part of the theoretical analysis aims to justify a key part of the design of LPM:

*Why do we use the error model $g_\phi$ to estimate the expected prediction error of the previous model?*

To answer this, we consider the simpler alternative of using a pointwise intrinsic reward without expectation: $r^{i,\mathrm{point}} := \log \mathrm{MSE}(\theta) - \log \mathrm{MSE}(\theta_D)$ for a single sampled parameter $\theta$. We show that using pointwise intrinsic reward breaks the monotone relationship between the intrinsic reward and the information gain, while our design using $g_\phi$ makes $r^i$ a monotone indicator of IG by adding a simple expectation operation upon the pointwise approach.

**Theorem 4.2** (Necessity of Expectation in Intrinsic Reward for Monotonicity). *Let $\theta \in \Theta$ be model parameters with prior $p(\theta)$ and posterior $p(\theta \mid D)$ given dataset $D$, and assume $\log p(D \mid \theta) = -c \log \mathrm{MSE}(\theta) + \mathrm{const}(D), c > 0$. Define simple pointwise intrinsic reward*

$$r^{i,point} := \log \mathrm{MSE}(\theta) - \log \mathrm{MSE}(\theta_D)$$

*for a single parameter $\theta$, and the expectation-based intrinsic reward*

$$r^{i,exp} := \mathbb{E}_{p(\theta)}[\log \mathrm{MSE}(\theta)] - \log \mathrm{MSE}(\theta_D),$$

*where $\theta_D$ is a chosen point in $\Theta$ satisfying $\log p(D \mid \theta_D) \geq \mathbb{E}_{p(\theta|D)}[\log p(D \mid \theta)]$. Then:*

1. *$r^{i,exp} \geq \frac{1}{c} \mathrm{IG}$, where IG is defined in Definition 1.*
2. *There exist $\theta$ for which $r^{i,point} < 0$ while $\mathrm{IG} > 0$.*

*Consequently, the expectation in the first term of $r_i$ is necessary to guarantee a deterministic monotone relationship between intrinsic reward and information gain.*

Proof is in Appendix A.4. Theorem 4.2 highlights the importance of using the expected prediction error (captured via the error model $g_\phi$) rather than a single-sample estimate. Without the expectation, the intrinsic reward can fluctuate and even become negative despite positive information gain, breaking the monotone correspondence. By maintaining a running estimate of the expected error, LPM ensures stable and reliable intrinsic rewards that faithfully track the model's learning progress.

In summary, our theoretical analysis justifies both the form of the LPM intrinsic reward and the use of an error model to estimate expected previous errors. While it is difficult to analyze the robustness of the above Theorems in the absence of the i.i.d. Gaussian observation model assumption, empirical experiments show that LPM consistently outperforms state-of-the-art baselines in various noisy environments, which we present in the following section.

## 5 EXPERIMENT

We designed and conducted the experiments to answer three key research questions. *In both environments with and without noisy-TVs,*

Table 1: Summary of baseline intrinsic motivation methods. $s_t$ and $s_{t+1}$ denote the current and next state, $a_t$ is the action, $\mathcal{M}$ represents episodic memory, and $d(\cdot, \cdot)$ is a distance metric.

| Method | Intrinsic Reward Function $r_t^i$ | Category |
|---|---|---|
| ICM (Pathak et al., 2017) | $\|\phi(s_{t+1}) - \hat{\phi}(s_{t+1}\|s_t, a_t)\|^2$ | Curiosity |
| Ensemble (Pathak et al., 2019) | $\mathrm{Var}_i[f_i(s_t, a_t)]$ | Curiosity |
| RND (Burda et al., 2018) | $\|f(s_{t+1}) - \hat{f}(s_{t+1})\|^2$ | Curiosity |
| AMA (Mavor-Parker et al., 2022) | $\|s_{t+1} - f(s_t, a_t)\|^2 - \lambda \Sigma(s_t, a_t)$ | Curiosity (Epistemic Estimation) |
| EME (Wang et al., 2024) | $d(s', s) \cdot \min\{\max\{\zeta(r_s), 1\}, M\}$ | Episodic Bonus (Metric-Based) |
| EDT (Jiang et al., 2025) | $\min_{s' \in \mathcal{M}} d(s_{t+1}, s')$ | Episodic Bonus (Similarity) |
| **LPM (Ours)** | $\mathbb{E}_{\mathcal{D}}\left[\varepsilon_t^{(\tau-1)}(o_{t+1})\right] - \varepsilon_t^{(\tau)}(o_{t+1})$ | **Learning Progress** |

(a) Noisy MNIST      (b) MiniWorld 3D Maze      (c) Multiple Atari games

Figure 1: Rendering of the experiment environments.

1. *Does the intrinsic reward of LPM converges faster compared with the existing methods while robust to aleatoric uncertainty?*
2. *In pure exploration tasks, does LPM explore more novel states in a given time duration compared with the existing methods?*
3. *In tasks with extrinsic reward, does LPM achieve higher extrinsic reward compared with the existing methods?*

To answer the above three research questions, we have designed three dedicated experiments, each for one research question. To evaluate the *scalability* of LPM, we used experiment environments with large observation space with increasing task complexity, from the noisy MNIST dataset (Mavor-Parker et al., 2022) to noisy Atari game RL environments (Mazzaglia et al., 2022; Wang et al., 2024). We also evaluate the robustness and consistency of all methods across deterministic and stochastic environmental conditions throughout the experiments. We compare LPM against state-of-the-art intrinsic motivation methods representing different algorithmic families. Table 1 summarizes the baselines, their intrinsic reward formulations, and the category they belong to.

From the experiments, we demonstrate that while existing intrinsic motivation methods encounter performance degration in stochastic environments, LPM consistently shows superior performance across both deterministic and stochastic settings. We now introduce the experiment settings, baselines, and results of each experiment.

## 5.1 NOISY MNIST

To answer the first research question, we evaluate our method using the noisy MNIST dataset following the experimental setup from (Mavor-Parker et al., 2022; Pathak et al., 2019; Mazzaglia et al., 2022).The noisy MNIST (Deng, 2012) environment contains separated deterministic and stochastic state transitions, allowing us to observe the behavior characteristic across different intrinsic motivation methods. The deterministic state starts with initial state of MNIST image of 0, and transit to next state that is exactly the same as initial state. While the stochastic state starts with initial state of MNIST image 1, and transit to randomly sampled next state of MNIST image 2-9. We focus on two representative baselines: AMA, a noise-robust curiosity-driven method, and Episodic Novelty Through Temporal Distance (EDT) (Jiang et al., 2025), an similarity based episodic bonus method. These two baselines serve as representatives of their respective families, providing a clear comparison of how different intrinsic motivation mechanisms behave as more transitions are observed.

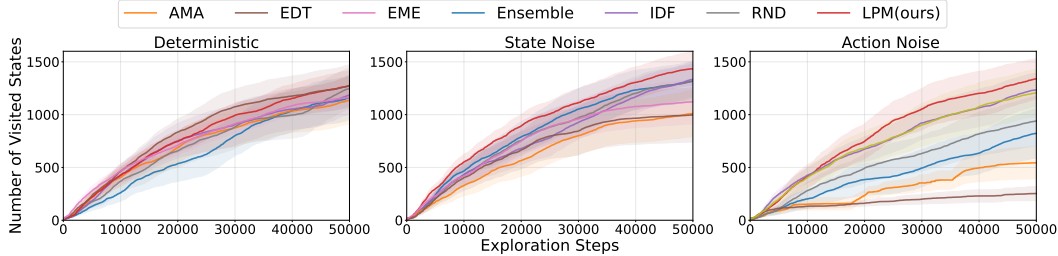

Figure 3: **Exploration performance across noise conditions, averaged over 10 random seeds.** State coverage during training in 3D maze environment. **Left:** Deterministic setting (best: EDT state coverage 1276.1). **Center:** State noise condition (best: LPM state coverage 1434.0). **Right:** Action noise setting (best: LPM state coverage 1340.0).

Figure 2 shows the shift of intrinsic motivation signal over training steps. Our LPM method provides consistent intrinsic rewards for both deterministic and stochastic transitions from the beginning, with both converging to zero after approximately 150 steps, representing immediate robustness to environmental stochasticity. AMA eventually converge to zero intrinsic reward for both transition types but requires significantly more exploration steps ($\approx 400$). Notably, AMA provides different reward magnitudes for deterministic versus stochastic transitions during training, meaning agents would temporarily favor stochastic states—indicating partial susceptibility to the noisy TV problem. Finally EDT, measuring similarity between next states, is incapable of overcome the stochasticity in transitions and keep finding the stochastic transitions more interesting.

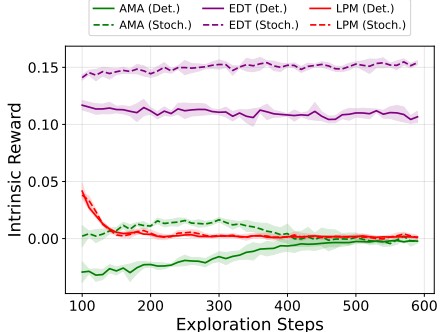

Figure 2: Noisy MNIST with deterministic and stochastic transitions over 5 runs.

## 5.2 MINIWORLD

To answer the second research question—whether LPM explores more novel states in pure exploration tasks compared with existing methods—we evaluate our method in the 3D MiniWorld environment (Chevalier-Boisvert et al., 2023) with high-dimensional observations (160×120 RGB). This experiment focuses on pure exploration performance, measuring the agent's ability to discover and visit diverse states when guided solely by intrinsic motivation methods. We use a 3-room N-shaped MiniWorld maze with 2,176 visitable states. Each room has distinct textures to increase visual diversity, making exploration visually complex and closer to real-world environments. Additionally, we test exploration consistency across different noise levels, challenging existing methods that may fail in realistic visual environments where unpredictable stimuli can trigger noisy-TV behavior by creating three experimental conditions to evaluate robustness: (1) a deterministic setting with fixed textures and transitions, (2) a state noise setting where the middle room exits a noisy wall displaying random pixel noise at each timestep to test whether agents fixate on unpredictable visual stimuli, and (3) an action noise setting where selecting the idle action replaces the true observation with a random CIFAR-10 image (Krizhevsky et al., 2009), simulating action-triggered noise. This follows similar noise setting as MiniGrid environment from (Mavor-Parker et al., 2022).

Figure 3 shows the exploration state coverage across three MiniWorld environments, averaged over 10 runs. Our LPM achieved the highest state coverage across all environments with 1,347.6 average visited states, outperforming the second-best method by 95.3 states (7.6% improvement in exploration efficiency). Notably, LPM demonstrates remarkable robustness to environmental perturbations, maintaining consistent performance across deterministic, state noise, and action noise conditions, while all baseline methods exhibit significant performance degradation in noisy environments—particularly under action noise. For similarity-based intrinsic motivation methods, this degradation in action noise environments can be attributed to the increased visual complexity introduced by noise (CIFAR images), which requires extensive data collection before agents can effec-

Table 2: State coverage comparison in the MountainCar continuous-control environment with deterministic and stochastic settings. Best performance in each task is in **bold**.

| Method | Deterministic Coverage (%) | Stochastic Coverage (%) | Drop (%) |
|---|---|---|---|
| LPM | $76.50 \pm 9.08$ | $\mathbf{67.04 \pm 14.60}$ | **12.4** |
| Ensemble | $\mathbf{91.22 \pm 2.04}$ | $61.02 \pm 5.03$ | 33.1 |
| RND | $45.50 \pm 14.53$ | $28.00 \pm 10.10$ | 38.5 |
| AMA | $33.00 \pm 12.31$ | $13.20 \pm 4.62$ | 60.0 |
| EDT | $82.16 \pm 13.57$ | $53.52 \pm 10.53$ | 34.9 |
| EME | $89.16 \pm 3.40$ | $32.46 \pm 11.31$ | 63.6 |
| IDF | $90.92 \pm 5.13$ | $12.80 \pm 3.19$ | 85.9 |

tively distinguish signal from stochasticity. This observation aligns with our earlier findings in the Noisy MNIST experiment.

### 5.3 MOUNTAIN CAR

To complement the MiniWorld experiment—which evaluates state coverage in a discrete action setting, we further conducted a continuous control exploration task. We constructed two variants of the MountainCar-Continuous environment to evaluate LPM under sparse rewards, continuous environments.

The deterministic environment includes the standard two-dimensional state (position, velocity) and one-dimensional continuous action. We remove all original rewards and place three reward points at random locations per run, creating a sparse-reward continuous control task. The stochastic environment extend the action space with an additional noisy action dimension. When this extended action is chosen with a value greater than zero, the environment freezes the true state and outputs a random observation sampled uniformly from [-1,1]. During testing, we separate the MountainCar state space into 100 bins and measure how many unique bins each method visits during training with 5 random seeds. As shown in Table 2, LPM can motivate exploration in deterministic continues environment, attains the best performance under stochasticity and shows the smallest degradation, with only a 12.4% drop, which is far lower than all baselines. This highlights LPM's strong robustness to stochasticity.

### 5.4 ATARI

To answer the third research question, we evaluate our method on several hard exploration Atari games identified in prior work (Wang et al., 2024; Bellemare et al., 2016; Wang et al., 2023), including Space Invader and Ms PacMan. To systematically test noise robustness while preserving meaningful gameplay, we extend each game's action space by adding two idle (no-op) actions. This preserves the utility of idle actions while allowing for controlled noise injection. For each game, we construct two environment variants: (1) the original Atari game without modifications, and (2) an action noise variant, where selecting an idle action replaces the true observation with a random CIFAR-10 image (Krizhevsky et al., 2009), following the noise-injection setup from (Mavor-Parker et al., 2022). This design allows us to directly evaluate how intrinsic motivation methods handle noise when exploring challenging, goal-directed tasks.

As shown in Figure 4, LPM achieves superior performance (4 out of 6) against strong baselines in Atari environment. More importantly, LPM demonstrates remarkable stability under noise, with only a 3.9% drop in Space Invader, 4.7% drop in UpNDown and even 0.3% improvement in Ms PacMan. In contrast, other methods show substantial vulnerability: EME, the strongest performer on clean Space Invader, completely fails under noise (100% drop), and all other baselines suffer varying degrees of performance loss. These results highlight that LPM provides the most consistent and noise-robust exploration strategy, making it especially suitable for real-world environments where unpredictable noise is common.

### 5.5 MONTEZUMA'S REVENGE

Beyond the standard Atari benchmarks, we further evaluate LPM on Montezuma's Revenge, one of the most challenging exploration games where most intrinsic-motivation methods fail to obtain any score. While prior experiments showed that LPM is highly robust to environmental stochasticity, this test examines whether LPM can still guide exploration in extremely sparse-reward settings. Remarkably in Figure 5, LPM is able to generate meaningful exploration behavior and achieves non-

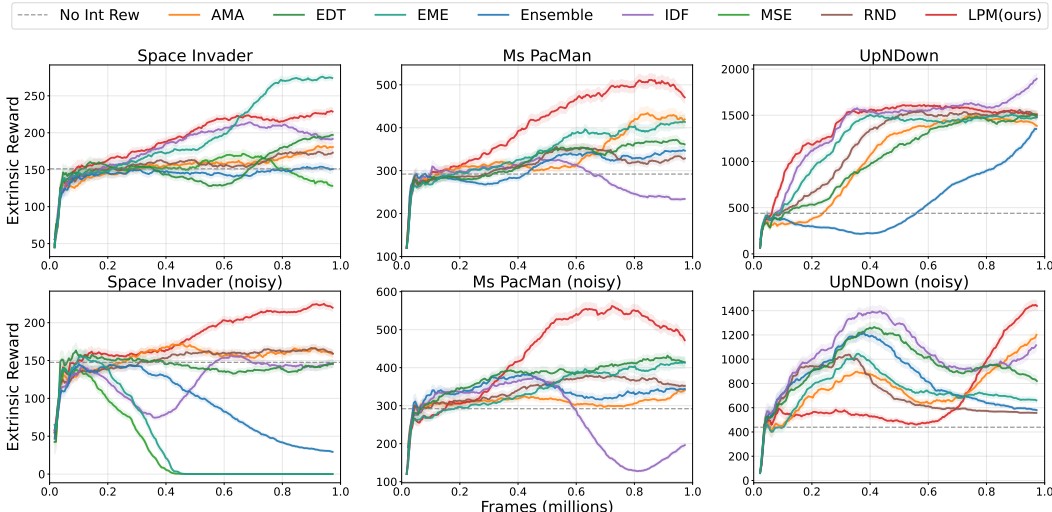

Figure 4: Average and standard deviation extrinsic rewards achieved by each method across deterministic and stochastic Atari environments with 128 random seeds.

trivial performance within 20M exploration steps, whereas RND requires 50M exploration steps. Since NGU(Badia et al., 2020) has no official implementation, we re-implemented it following the original paper. NGU requires days of wall-clock training, so we report only 50M steps, during which it failed to obtain any extrinsic reward. This result suggests that LPM is not only noise-robust but also capable of driving exploration in hard long-horizon tasks such as Montezuma's Revenge.

## 5.6 COMPUTATIONAL OVERHEAD

While LPM employs a dual-network structure to estimate learning progress, its computational complexity remains modest when compared to baseline methods such as AMA (Mavor-Parker et al., 2022), Ensemble (Pathak et al., 2019), and EME (Wang et al., 2024). AMA uses a double-headed neural network, which require similar computational cost with LPM. Ensemble and EME both require multiple models and hence require significantly higher computational costs.

## 6 RELATED WORK

### 6.1 CURIOSITY-DRIVEN EXPLORATION

Curiosity-driven exploration methods use the prediction error of a learned dynamics model as intrinsic reward, encouraging the agent to visit states where its predictions are inaccurate (Schmidhuber, 1991; Burda et al., 2019). However, this approach can fail catastrophically in stochastic environments, where prediction errors may be dominated by uncontrollable randomness rather than meaningful novelty. To mitigate this issue, the inverse dynamics feature (IDF) curiosity model (Pathak et al., 2017) computes prediction errors only in a feature space that captures aspects of the environment controllable by the agent, reducing sensitivity to irrelevant stochasticity. Random Network Distillation (RND) (Burda et al., 2018) instead defines intrinsic rewards using the prediction error of a network trained to match the output of a fixed, randomly initialized target network. Other approaches use ensemble methods, where intrinsic rewards are based on the disagreement between predictions from multiple dynamics models (Pathak et al., 2019). Aleatoric Mapping Agent (AMA) (Mavor-Parker et al., 2022) explicitly filter out aleatoric uncertainty. While these methods have made important progress, sample efficiency still remains as a major challenge. Accurately disentangling epistemic uncertainty from aleatoric uncertainty generally requires extensive data collection (An et al., 2025). During the early stages of training, these approaches often provide unreliable intrinsic rewards dominated by noise, wasting samples and slowing the agent's ability to explore meaningfully in complex environments.

### 6.2 EXPLORATION THROUGH EPISODIC BONUS

Episodic bonus methods provide intrinsic motivation by rewarding state novelty within the current episode (Zhang et al., 2021; Henaff et al., 2023). These approaches generally fall into two categories:

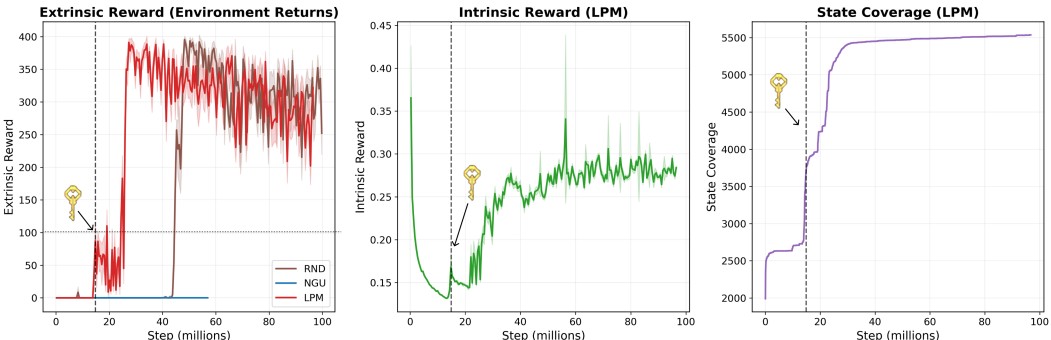

Figure 5: Exploration performance on Montezuma's Revenge. LPM enables the agent to achieve meaningful extrinsic reward within 20M steps, whereas baselines require 40M+ steps to reach comparable scores. **Center:** LPM's intrinsic reward gradually decays as exploration saturates, mirroring its behavior on Noisy MNIST. Once the agent enters a previously unseen region around 20M steps, the intrinsic reward spikes, guiding the agent toward high-reward states. **Right:** LPM steadily expands its state coverage and discovers novel states demonstrating its ability to drive exploration even in extremely sparse-reward environments.

*Count-based methods* (Raileanu & Rocktäschel, 2020; Flet-Berliac et al., 2021; Mu et al., 2022; Zhang et al., 2021; Kapturowski et al., 2024) generate a positive intrinsic reward signal whenever the agent visits a previously unseen state within an episode, thereby encouraging exploration of novel states. In contrast, *similarity-based methods* (Savinov et al., 2018; Wang et al., 2024; Jiang et al., 2025; Yang et al., 2024) define novelty as a continuous function of the distance between the current state and past visited states, using metrics such as temporal distance (Jiang et al., 2025), Kullback–Leibler divergence (Wang et al., 2024), or Euclidean distance (Henaff et al., 2022).

While count-based approaches are simple and avoid repeatedly rewarding duplicate visits, they are computation inefficient in high-dimensional stochastic environments. Similarity-based episodic bonuses reduce some of these memory requirements by generalizing across states via distance metrics. However, this depends on collecting a sufficiently diverse episodic memory before they can reliably escape stochasticity. During early exploration, when memory is sparse, these methods tend to assign high intrinsic rewards to stochastic or uninformative states, wasting large numbers of samples and slowing the discovery of meaningful, controllable transitions in complex environments.

### 6.3 INFORMATION-THEORETIC EXPLORATION

Information-theoretic exploration methods reward *information gain*, encouraging the agent to take actions that maximally reduce its uncertainty about the environment dynamics (Houthooft et al., 2016; Rhinehart et al., 2021). This is typically formalized as information gain (IG), the KL divergence between the agent's posterior and prior beliefs upon receiving a new observation. By directly maximizing IG, these approaches have strong theoretical guarantees that the collected data will be maximally informative for learning state transitions. In practice, estimating information gain requires maintaining a posterior distribution over the environment, which is typically achieved using Gaussian processes (GPs) (Bottero et al., 2022; Sui et al., 2015; Hennig & Schuler, 2012; Hernández-Lobato et al., 2014) or Bayesian neural networks (BNNs) (Houthooft et al., 2016; Mazzaglia et al., 2022; Blau et al., 2019; Li & Faisal, 2021). While theoretically appealing, these methods are computationally expensive and hard to scale to high-dimensional, visual domains such as Atari or robotics.

## 7 CONCLUSION

Motivated by recent neuroscience findings (Ten et al., 2021), we propose using learning progress as intrinsic reward to motivate reinforcement learning exploration. We designed the Learning Progress Monitoring (LPM) method with a dual network structure to monitor learning progress. We then thoroughly tested our method in environments with and without extrinsic rewards, state-based noise, and action-based noise. Our results demonstrate that learning progress as intrinsic reward, compared to existing intrinsic motivation methods that use prediction error or distribution similarity, achieves higher exploration efficiency while being naturally robust to noise. This makes our LPM method particularly suitable for realistic environments where deterministic and stochastic elements coexist.

## ETHICS STATEMENT

All authors have read and adhere to the ICLR Code of Ethics.

## REPRODUCIBILITY STATEMENT

Detailed experimental setups, hyperparameters, architectural specifications, and links to our source code are documented in Appendix C.

## ACKNOWLEDGMENT

This work was supported in part by the NSF Grant #2239458 and an UC Merced Fall 2023 Climate Action Seed Competition Grant. Any opinions, findings, and conclusions expressed in this material are those of the authors and do not necessarily reflect the views of the funding agencies.

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

## A  PROOFS

### A.1  DERIVATION OF INFORMATION GAIN (IG) TO KULLBACK–LEIBLER (KL) DIVERGENCE

Here we show that the Definition 1 of IG is equivalent to the Kullback–Leibler (KL) divergence from the posterior to the prior:

$$IG = KL(p(\theta \mid D) \,\|\, p(\theta)) \tag{4}$$

$$= \int p(\theta \mid D) \, \log \frac{p(\theta \mid D)}{p(\theta)} \, d\theta \tag{5}$$

$$= \int p(\theta \mid D) \Big[ \log p(\theta \mid D) - \log p(\theta) \Big] \, d\theta \tag{6}$$

$$= \int p(\theta \mid D) \Big[ \log p(D \mid \theta) + \log p(\theta) - \log p(D) - \log p(\theta) \Big] \, d\theta \tag{7}$$

$$= \int p(\theta \mid D) \, \log p(D \mid \theta) \, d\theta \; - \; \log p(D) \tag{8}$$

$$= \mathbb{E}_{p(\theta|D)} \big[ \log p(D \mid \theta) \big] \; - \; \log p(D). \tag{9}$$

### A.2  MLE SATISFIES THE $\theta_D$ CONDITION

**Lemma A.1.** *Let $\theta_{\mathrm{MLE}} = \arg\max_\theta \log p(D \mid \theta)$. Then*

$$\log p(D \mid \theta_{\mathrm{MLE}}) \; \geq \; \mathbb{E}_{p(\theta|D)}[\log p(D \mid \theta)].$$

*Proof.* The expectation is an average of log-likelihood values, which cannot exceed their maximum. □

## A.3 PROOF OF THEOREM 4.1

Reiterating the theorem:

**Theorem A.2** (Monotonicity and Zero Equivalence). *Let $\theta \in \Theta$ be model parameters with prior $p(\theta)$ and posterior $p(\theta \mid D)$ given dataset $D$. Assume the likelihood depends on $\theta$ only through a positive scalar function $\mathrm{MSE}(\theta)$ such that $\log p(D \mid \theta) = -c \log \mathrm{MSE}(\theta) + \mathrm{const}(D), c > 0$. For intrinsic reward $r^i := \mathbb{E}_{p(\theta)}[\log \mathrm{MSE}(\theta)] - \log \mathrm{MSE}(\theta_D)$ where $\theta_D$ is a chosen point in $\Theta$ satisfying $\log p(D \mid \theta_D) \geq \mathbb{E}_{p(\theta \mid D)}[\log p(D \mid \theta)]$, Then the following hold:*

1. *$r_i \geq \frac{1}{c} \mathrm{IG}$, where $\mathrm{IG}$ is defined by Definition 1.*
2. *$\mathrm{IG} = 0 \implies r_i = 0$.*
3. *$r_i = 0 \implies \mathrm{IG} = 0$ under the identifiability condition that the likelihood is non-constant and injective in $\mathrm{MSE}(\theta)$.*

*Proof.* From the likelihood assumption,

$$\log \mathrm{MSE}(\theta) = -\frac{1}{c}\big(\log p(D \mid \theta) - \mathrm{const}(D)\big).$$

Hence

$$r_i = \mathbb{E}_{p(\theta)}[\log \mathrm{MSE}(\theta)] - \log \mathrm{MSE}(\theta_D) = \frac{1}{c}\Big(\log p(D \mid \theta_D) - \mathbb{E}_{p(\theta)}[\log p(D \mid \theta)]\Big).$$

By Jensen's inequality,

$$\log p(D) = \log \int p(D \mid \theta)p(\theta)\, d\theta \geq \mathbb{E}_{p(\theta)}[\log p(D \mid \theta)].$$

Combined with the condition $\log p(D \mid \theta_D) \geq \mathbb{E}_{p(\theta \mid D)}[\log p(D \mid \theta)]$ and the information gain identity

$$\mathrm{IG} = \mathbb{E}_{p(\theta \mid D)}[\log p(D \mid \theta)] - \log p(D),$$

we obtain

$$r_i \geq \frac{1}{c}\big(\mathbb{E}_{p(\theta \mid D)}[\log p(D \mid \theta)] - \log p(D)\big) = \frac{1}{c}\mathrm{IG}.$$

If $\mathrm{IG} = 0$, then $p(\theta \mid D) = p(\theta)$ almost surely, implying $\mathrm{MSE}(\theta)$ is constant over $\theta$. Hence $r_i = 0$.

Conversely, if $r_i = 0$ and the likelihood is non-constant and injective in $\mathrm{MSE}(\theta)$, then $\log \mathrm{MSE}(\theta)$ is constant under $p(\theta)$, forcing $p(\theta \mid D) = p(\theta)$, so $\mathrm{IG} = 0$. □

**Corollary A.2.1** (Gaussian i.i.d. Case with MLE). *Let $D = \{(x_i, y_i)\}_{i=1}^n$ and assume a Gaussian i.i.d. residual model*

$$y_i = f(x_i; \theta) + \varepsilon_i, \qquad \varepsilon_i \sim \mathcal{N}(0, \mathrm{MSE}(\theta)),$$

*so that the log-likelihood is*

$$\log p(D \mid \theta) = -\frac{n}{2} \log \mathrm{MSE}(\theta) + \mathrm{const}(D).$$

*Let $\theta_D$ be the maximum likelihood estimate (MLE):*

$$\theta_D = \arg\max_\theta \log p(D \mid \theta).$$

*Then the intrinsic reward*

$$r_i := \mathbb{E}_{p(\theta)}[\log \mathrm{MSE}(\theta)] - \log \mathrm{MSE}(\theta_D)$$

*satisfies*

$$r_i \geq \frac{2}{n} \mathrm{IG}, \qquad \mathrm{IG} := \mathrm{KL}(p(\theta \mid D)\|p(\theta)).$$

*Moreover, $\mathrm{IG} = 0 \implies r_i = 0$ and the converse holds under the identifiability condition that the likelihood is injective in $\mathrm{MSE}(\theta)$.*

*Proof.* Immediate from Theorem 1 with $c = n/2$ and the MLE choice of $\theta_D$, which guarantees

$$\log p(D \mid \theta_D) \geq \mathbb{E}_{p(\theta \mid D)}[\log p(D \mid \theta)].$$

Substituting $c = n/2$ into the lower bound of Theorem 1 gives $r_i \geq \frac{2}{n}\mathrm{IG}$. □

A.4    PROOF OF THEOREM 4.2

Reiterating the theorem:

**Theorem A.3** (Necessity of Expectation in Intrinsic Reward for Monotonicity). *Let $\theta \in \Theta$ be model parameters with prior $p(\theta)$ and posterior $p(\theta \mid D)$ given dataset $D$, and assume $\log p(D \mid \theta) = -c \log \text{MSE}(\theta) + \text{const}(D), c > 0$. Define simple pointwise intrinsic reward*

$$r^{i,point} := \log \text{MSE}(\theta) - \log \text{MSE}(\theta_D)$$

*for a single parameter $\theta$, and the expectation-based intrinsic reward*

$$r^{i,exp} := \mathbb{E}_{p(\theta)}[\log \text{MSE}(\theta)] - \log \text{MSE}(\theta_D),$$

*where $\theta_D$ is a chosen point in $\Theta$ satisfying $\log p(D \mid \theta_D) \geq \mathbb{E}_{p(\theta|D)}[\log p(D \mid \theta)]$.*

*Then:*

1. *$r^{i,exp} \geq \frac{1}{c} \text{IG}$, where $\text{IG} = \text{KL}(p(\theta \mid D)\|p(\theta))$.*
2. *There exist $\theta$ for which $r^{i,point} < 0$ while $\text{IG} > 0$.*

*Consequently, the expectation in the first term of $r_i$ is necessary to guarantee a deterministic monotone relationship between intrinsic reward and information gain.*

*Proof.* Express each reward in terms of the log-likelihood:

$$\log \text{MSE}(\theta) = -\frac{1}{c}(\log p(D \mid \theta) - \text{const}(D)),$$

so that

$$r^{i,point} = \frac{1}{c}\big(\log p(D \mid \theta_D) - \log p(D \mid \theta)\big), \quad r^{i,exp} = \frac{1}{c}\big(\log p(D \mid \theta_D) - \mathbb{E}_{p(\theta)}[\log p(D \mid \theta)]\big).$$

For $r^{i,exp}$, Jensen's inequality gives

$$\log p(D) = \log \int p(D \mid \theta)p(\theta)d\theta \geq \mathbb{E}_{p(\theta)}[\log p(D \mid \theta)].$$

Combined with $\log p(D \mid \theta_D) \geq \mathbb{E}_{p(\theta|D)}[\log p(D \mid \theta)]$, we get

$$r^{i,exp} \geq \frac{1}{c}\big(\mathbb{E}_{p(\theta|D)}[\log p(D \mid \theta)] - \log p(D)\big) = \frac{1}{c}\text{IG}.$$

For $r^{i,point}$, however, consider any $\theta$ for which $\log p(D \mid \theta) > \log p(D \mid \theta_D)$ (possible because $\theta$ may be drawn from the prior or outside high-likelihood regions). Then

$$r^{i,point} = \frac{1}{c}(\log p(D \mid \theta_D) - \log p(D \mid \theta)) < 0$$

while $\text{IG} = \text{KL}(p(\theta \mid D)\|p(\theta)) \geq 0$. Therefore, monotonicity fails without the expectation. This demonstrates the necessity of using the expectation in the first term of $r^i$.    □

## B    MINIGRID EXPERIMENT

We further evaluate LPM within the MiniGrid(Chevalier-Boisvert et al., 2023) framework, using the Lava Crossing environment. This maze-like domain serves as an ideal testbed for intrinsic motivation: while agents must explore to reach the goal, excessive or unregulated curiosity can be detrimental, driving the agent into fatal lava grids. As illustrated in Figure 6, both ICM and our proposed LPM demonstrate comparable performance, significantly outperforming the remaining baselines.

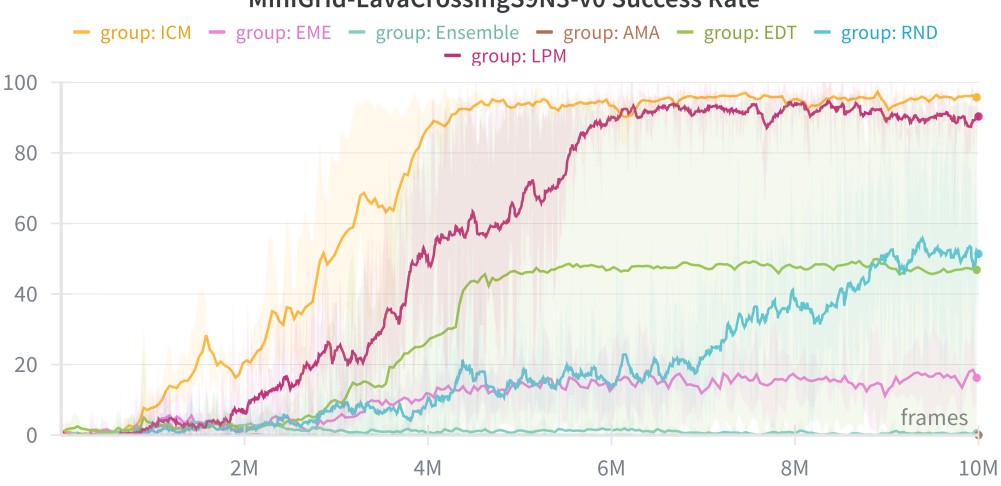

Figure 6: PPO Agent success rate across 5 random seeds.

## C    IMPLEMENTATION DETAILS

This appendix provides comprehensive implementation details for all experiments to ensure full reproducibility of the reported results. We present the specific network architectures, hyperparameters, and training configurations used in each experimental setting.

### C.1    NOISY MNIST IMPLEMENTATION DETAILS

**Network Architectures**

**LPM Dynamic:**

- Layer 1: Linear(784 → 784) → ReLU
- Layer 2: Linear(784 → 784) → ReLU
- Layer 3: Linear(784 → 784) → ReLU
- Concatenation: Concat(input, features)
- Output Layer: Linear(1568 → 784)

**LPM Error Model:**

- Layer 1: Linear(784 → 256) → ReLU
- Layer 2: Linear(256 → 128) → ReLU
- Layer 3: Linear(128 → 64) → ReLU
- Output Layer: Linear(64 → 1)

**Training Configuration**

- Learning rate: 0.001
- Buffer size 100, update every 1 step
- Batch size: 32
- Optimizer: Adam
- Total training steps: 600
- Evaluation frequency: Every 10 steps
- Random seeds: 5 seeds for statistical significance

## C.2 MINIWORLD IMPLEMENTATION DETAILS

All methods use identical CNN architecture to ensure fair comparison:

- **Conv1:** Conv2d(3 → 32, kernel=8, stride=4) → ReLU
- **Conv2:** Conv2d(32 → 64, kernel=4, stride=2) → ReLU
- **Conv3:** Conv2d(64 → 64, kernel=3, stride=1) → ReLU
- **Flatten:** Reshape to feature vector

**LPM-Specific Architecture**

**Dynamics Model ($f_\theta$):**

- Input: CNN Features
- Layer 1: Linear(features+actions → 512) → ReLU
- Decoder: ConvTranspose Decoder → Sigmoid

**Error Model ($g_\phi$):**

- Input: CNN Features
- Layer 1: Linear(features+actions → 256) → ReLU
- Layer 2: Linear(256 → 128) → ReLU
- Output Layer: Linear(128 → 1)

**Training Configuration**

**Shared RL Parameters (All Methods):**

- Base Algorithm: A2C
- Policy Learning Rate: 0.01
- Discount Factor ($\gamma$): 0.99
- Intrinsic Weight ($\lambda$): 1
- Entropy Coefficient: 0.03
- Value Loss Coefficient: 0.5
- Gradient Clipping: 0.5
- Update Frequency: 64 steps
- Episodes per Method: 10
- Steps per Episode: 50,000
- Dynamics Model Learning Rate: 0.001

**LPM-Specific Parameters:**

- Error Buffer Size: 100

## C.3 C.3 ATARI IMPLEMENTATION DETAILS

**Shared CNN Feature Extractor**

All methods use the same CNN architecture for fair comparison:

- **Conv1:** Conv2d(4 → 32, kernel=8, stride=4) → LeakyReLU
- **Conv2:** Conv2d(32 → 64, kernel=4, stride=2) → LeakyReLU
- **Conv3:** Conv2d(64 → 64, kernel=3, stride=1) → LeakyReLU
- **Flatten:** Reshape to feature vector

- **FC:** Linear(conv_output $\rightarrow$ 512) $\rightarrow$ LeakyReLU

**LPM-Specific Architecture**

**LPM Dynamics Model ($f_\theta$):**

- Input: CNN Features
- Layer 1: Linear(512+actions $\rightarrow$ 512) $\rightarrow$ ReLU
- Decoder: ConvTranspose Decoder $\rightarrow$ Sigmoid

**LPM Error Model ($g_\phi$):**

- Input: CNN Features
- Layer 1: Linear(512+actions $\rightarrow$ 256) $\rightarrow$ ReLU
- Layer 2: Linear(256 $\rightarrow$ 128) $\rightarrow$ ReLU
- Output Layer: Linear(128 $\rightarrow$ 1)

**Training Configuration**

**Shared RL Parameters (All Methods):**

- Base Algorithm: PPO Schulman et al. (2017)
- Learning Rate: 1e-4
- Clip Parameter: 0.1
- PPO Epochs: 3
- Mini-batches: 8
- Entropy Coefficient: 0.001
- Value Loss Coefficient: 0.5
- Discount Factor ($\gamma$): 0.99
- GAE: True
- Processes: 64
- Steps per Update: 128

**LPM-Specific Parameters:**

- Buffer Size: 128

## C.4 COMPUTING INFRASTRUCTURE

All experiments were conducted on Google Colab Pro with the following specifications:

**Hardware:**

- GPU: NVIDIA Tesla T4 (16GB VRAM), NVIDIA A100 (40GB VRAM), NVIDIA RTX Pro 6000 Blackwell (96GB VRAM)
- CPU: High-RAM runtime (51GB)

## C.5 CODE AVAILABILITY

All source code for reproducing the experiments reported in this paper is available at `https://github.com/Akuna23Matata/LPM_exploration`

