# OpenReview forum: "Beyond Noisy-TVs: Noise-Robust Exploration Via Learning Progress Monitoring"
_ICLR.cc/2026/Conference — ICLR 2026 Poster_

### Official Review · Reviewer_vQEu · 2025-10-23

**Soundness:** 2
**Presentation:** 3
**Contribution:** 2
**Rating:** 4
**Confidence:** 3

**Summary:**

The paper presents the design of a new intrinsic reward based on maximizing a proxy for information gain under a learned model. The authors show that by rewarding the agent in consecutive iterations according to its capability to generate data which minimizes the error of the learned model, this leads to directionally optimizing for information gain under the model and to explorative agent behavior that achieves good performance. The main contribution is the theoretical proof which shows that the proposed intrinsic reward serves a directionally correct proxy for information gain under a learned model, which is a relevant and popular objective. The empirical evidence shows that the proposed method achieves comparable and sometimes superior performance than a subset of exploration baselines.

**Strengths:**

The motivation of the work and contribution (especially the theoretical contribution) is sound and rigorous. Information gain is a popular objecitve to optimize in model-based RL and particularly for enhanced exploration in intrinsically-motivated RL, and the presented intrinsic reward (which is very simple to understand and implement) is shown to be a correct directional estimator of the original and more complex information gain.

The paper is clearly written, rigorous and easy to follow. The evaluation protocols are described in detail and a great number of independent seeds is used to draw statistically significant conclusions.

**Weaknesses:**

The main weakness of the paper is the empirical evidence provided. While the evaluation protocols are rigoroous in some regards (e.g. a great number of indpendent seeds is used to aggregate results), the choice of environments is non-standard, and other training protocols (e.g. RL algorithm used, training steps, and choice of baselines) is not adequate.

 I think the paper would benefit from the following points:

- First evaluation of the method in classical exploration environments where the properties of the method can be understood in a perfectly controlled environment (e.g. like a fully-observed MDP). These can be gridworlds, or other popular exploration tasks like DeepSea.

- Addition of tasks that truly test for exploration capabilities. At the current stage, I don't think the results in Atari are convincing or relevant. The paper describes SpaceInvaders and MsPacman as "hard exploration environments" but they are really not, since they both provide dense rewards and are fully-observed. Additionally, the agents are only trained for 1M environment, steps which is extremely far from the well-tested standards in Atari of 200M. The performance of the algorithm is unknown beyond 1M environment steps, which weakens the claims. I believe the suite of tasks from MiniGrid, ProcGen, or Crafter would represent much more interesting evaluation suites.

- While there is an appropriate number of baselines used in the paper, it is unclear why those were chosen before others that are potentially more relevant and/or recent. e.g. E3B [1], Disagreement [2], NGU [3]. In general, the evaluation protocols of this work are not close or similar to other standards previously proposed in intrinsically-motivated RL [4].

- Related to the previous point, I think the authors could elaborate more in covering related work in intrinsically-motivated RL, especially recent algorithms that achieve great performance in noisy environments [5] and others based on active inference theory for RL [6,7,8,9], which is all about training agents to minimize the error/surprise of their internal world models (hence very relevant to this work), and is currently omitted in the paper. Since the authors cover curiosity and episodic methods, I would also suggest covering the combination of both [10,4].

With this, I think the paper presents an interesting and promising idea, and that the theoretical contribution is relevant. However, the poor evaluation and empirical evidence weakens the claims of the paper and should be improved and made more general to render this paper publishable.

[1] Henaff, Mikael, et al. "Exploration via elliptical episodic bonuses." Advances in Neural Information Processing Systems 35 (2022): 37631-37646.

[2] Pathak, Deepak, Dhiraj Gandhi, and Abhinav Gupta. "Self-supervised exploration via disagreement." International conference on machine learning. PMLR, 2019.

[3] Badia, Adrià Puigdomènech, et al. "Never give up: Learning directed exploration strategies." arXiv preprint arXiv:2002.06038 (2020).

[4] Yuan, Mingqi, et al. "Rlexplore: Accelerating research in intrinsically-motivated reinforcement learning." arXiv preprint arXiv:2405.19548 (2024).

[5] Saade, Alaa, et al. "Unlocking the power of representations in long-term novelty-based exploration." arXiv preprint arXiv:2305.01521 (2023).

[6] Berseth, Glen, et al. "SMiRL: Surprise minimizing RL in Entropic environments." (2019).

[7] Friston, Karl. "The free-energy principle: a unified brain theory?." Nature reviews neuroscience 11.2 (2010): 127-138.

[8] Rhinehart, Nicholas, et al. "Information is power: Intrinsic control via information capture." Advances in Neural Information Processing Systems 34 (2021): 10745-10758.

[9] Hugessen, Adriana, et al. "Surprise-Adaptive Intrinsic Motivation for Unsupervised Reinforcement Learning." arXiv preprint arXiv:2405.17243 (2024).

[10] Henaff, Mikael, Minqi Jiang, and Roberta Raileanu. "A study of global and episodic bonuses for exploration in contextual mdps." International Conference on Machine Learning. PMLR, 2023.

**Questions:**

Is the $g_{\phi}$ model reinitialized at each iteration to only fit the most recent model errors? or is it continually trained with the evolving world model?

---

> ### Author Response · Authors · 2025-11-20
> **Author rebuttal (I)**
>
> Thank you for your detailed and constructive feedback. Below we address the mentioned weakness:
>
> W1: classical exploration environments
>
> Existing works often use MiniGrid as their first experiment to study their method’s behavior and their exploration results as a controlled environment. However, this environment is too simplistic for studying this topic at this time: the observation dimension is too small to show the method’s applicability to real-world problems. We consider our experiments an upgrade from MiniGrid in this aspect.
>
> In our experiment design, we deliberately designed two experiments: First we use the fully controlled Noisy MNIST experiment to study the intrinsic motivation change as more state transitions are observed. This can be considered as a better version of MiniGrid as we remove the randomness of the RL agent and focus solely on the intrinsic motivation agent. Next we study in MiniWorld, which is considered as an upgraded, 3D version of MiniGrid, while preserving environment complexity, discrete action space, and sparse reward (we have zero reward in environment), we introduce more visual complexity, making it even harder for the models to differentiate meaningful states from stochasticity.
>
> W2: Hard exploration task
>
> As our paper focus on “efficient exploration while robust to noise”, we dedicatedly design these experiment to evaluate noise robustness. We set our training step enough to show that all baseline are affected in noisy Atari games while our LPM method managed to maintain its game play performance regardless of noise.
>
> In addition, we have added an additional experiment section to test our performance on Montezuma’s revenge, which is considered as the most classic hard exploration Atari game. We have tested our method with 50M steps, which is a sweet spot, that is more than enough for agent to acquire all scores in room 1 and 2. Surprisingly, our LPM method is the first method to acquire all scores in room 1 and 2 for a total of 400 scores within 20M steps, showing superiority  in terms of performance compared to RND and NGU. We point to the newly added Section 5.5 for details about this new experiment.
>
> W3: Choose of Baselines
>
> First, we would like to clarify that we did include “Self-Supervised Exploration via Disagreement” as our baseline; in table 1, we mistakenly cited the paper that proposed ensemble method instead of citing the correct disagreement paper that use variance of ensembled models to encourage exploration. This has been fixed.
>
> Due to the limitation of resources, we can only select a limited number of baselines; as a result, we have selected the latest (which, in their study, showed superiority compared to the earlier works). Specifically: we recognize NGU as one of the earliest paper to use episodic bonus to encourage exploration and E3B is also one of the famous paper that use a similarity based method for episodic bonus. Instead of them we included ETD, which is the latest work and is also a similarity based episodic bonus method just like E3B and NGU. All the earlier baselines (E3B, NGU, DEIR and etc) are studied in detail and compared against ETD, and ETD acquires best performance.
>
> W4: Related works
>
> We appreciate your suggestions and have carefully reviewed all the cited literature. While these works provide valuable perspectives, they still fall under the intrinsic-motivation categories already outlined in Section 6. We already include [2] as one of our baselines, and we have additionally incorporated [3] into our new Montezuma’s Revenge experiments. Several of the recommended works are also now cited in our related work section.
>
> Q1: The error prediction model continually trained with the evolving dynamic model on recently generated prediction errors stored in the buffer. We have detailed workflow in Algorithm 1, and model details in Appendix B.
>
> We thank you again for your insightful comments. Your suggestions helped us strengthen our experimental evaluation, and we have incorporated two additional experiments accordingly.

---

> > ### Comment · Reviewer_vQEu · 2025-11-26
> >
> > Thank you for your response, I appreciate the efforts in improving the paper. However, I am still not convinced by the empirical evidence provided or your arguments to support it.
> >
> > Again, exploration results with 1M env steps in Atari are not generalizable, and don't provide significant conclusions as many other algorithmic confounding factors could affect the early performance of different methods. The choice of Atari games is also arbitrary. I think adding Montezuma is a good idea, but the presented results are not very convincing (PPO+RND should achieve [better performance](https://wandb.ai/openrlbenchmark/openrlbenchmark/reports/-MontezumaRevenge-CleanRL-s-PPO-RND--VmlldzoyNTIyNjc5) than your reported results).
> >
> > I understand your argument to justify MetaWorld and NoisyMNIST, but I don't think it's enough to discard the addition of classic environments or MiniGrid (which provides a large suite of tasks which can be used to make claims more general if an algorithm dominates across the majority of tasks).
> >
> > Overall, I just find the provided empirical evidence does not currently meet the standards of quality and significance for ICLR.

---

> > > ### Author Response · Authors · 2025-12-03
> > > **Author Rebuttal (II)**
> > >
> > > Thank you for your feedback. We are confident that our experiment design is robust, self-consistent, and specifically tailored to validate our core claims of the paper: robustness to stochasticity, and theoretical validity. We again clarify regarding the reviewers’ concerns.
> > >
> > > **First, the 1M step horizon is deliberately chosen to measure *relative performance degradation* under noise, following established protocols for noise-robustness research.** Our protocol aligns directly with (How to Stay Curious while Avoiding Aleatoric Uncertainty from ICML 2022), who established this 1M steps benchmark on Atari games Bank Heist and Mario to specifically study noise robustness. By adhering to this precedent, we ensure our noise-robustness claims are comparable to the specific literature we aim to improve upon. Our objective is not to maximize scores in a noiseless setting (where 10 to 400M steps are commonly used), but to isolate the specific failure modes of intrinsic motivation methods under noise. In our experiments, baselines exhibit performance degradation early in training when action-based noise is present. And beyond studying the performance degradation under noise, we have added Montezuma’s Revenge during rebuttal period, as the most well-recognized hard exploration Atari game in 100M steps to study the long term exploration efficiency. Combining these experiments make our design comprehensive and confirm our claim.
> > >
> > > **Second, our reported PPO+RND baseline performance is better than the performance  of PPO+RND from CleanRL that you referenced.** Our implementation of PPO+RND found the first key at **~50M steps**, which is actually *faster* than the **~80M** steps reported in the CleanRL benchmarks you referenced. By carefully reviewed the code base of your linked benchmark, we found several implementation differences that may cause the reported performance different from our implementation therefore these reported statistics should not be used to compare against each other.
> > >
> > > - Environment version: We use **NoFrameskip-v4** , whereas CleanRL benchmarks use **v5** (which includes sticky actions and frameskip), which fundamentally changes the agent control.
> > > - Hyperparameters and objective: Your cited CleanRL’s implementation use an extrinsic coefficient (ext_coef) of **2.0** and an intrinsic discount of **0.99**. This biases the agent toward extrinsic rewards and suppresses long-term intrinsic exploration. Whereas we use extrinsic coefficient (ext_coef) of **1.0** and **no discount** for intrinsic motivation. This setup is deliberately chosen to study the impact of the intrinsic motivation algorithm.
> > >
> > > Our evaluation makes sure all baselines use the same environment and hyperparameter settings as reported in Appendix C.
> > >
> > > **Third, to address your request, we added the *MiniGrid-LavaCrossing* experiment (Appendix B), demonstrating that LPM drives efficient exploration.** As shown in Figure 6 and Appendix B, LPM matches the success rate of ICM and significantly outperforms all other baselines (EME, Ensemble (disagreement), AMA, EDT, RND). However we insist that MiniWorld serves as a challenging 3D evolution of MiniGrid, preserving the critical challenges of discrete actions and sparse rewards while introducing high-dimensional visual complexity that makes differentiating meaningful states from stochasticity significantly harder.
> > >
> > > In summary, our experiment includes **Noisy MNIST, MiniGrid, 3D MiniWorld, Continuous MountainCar, Atari, and Montezuma’s Revenge**. This diverse set of environments rigorously tests our method across the full spectrum of RL environments: from varying **visual complexities** (simple grids to high-dimensional pixel images) to continuous vs. discrete action spaces, and sparse vs. rewardless settings. This comprehensive empirical evidence validates our core hypothesis: LPM is both theoretically sound and empirically robust, demonstrating the **least performance degradation** under noise while achieving promising **exploration efficiency** in hard exploration tasks like Montezuma’s Revenge.

---

### Official Review · Reviewer_Hppm · 2025-10-24

**Soundness:** 3
**Presentation:** 2
**Contribution:** 3
**Rating:** 8
**Confidence:** 3

**Summary:**

The paper studies how intrinsic motivation methods in RL fail in stochastic environments, the classic “noisy TV” problem. It proposes Learning Progress Monitoring (LPM), which rewards how much a dynamics model improves rather than how much error it has. The key idea is to compare the current log-prediction error to an expected previous error estimated by a small network. The authors show that this bonus can be viewed as a monotone, noise-robust approximation of information gain. Experiments on Noisy-MNIST, MiniWorld with injected noise, and Atari with action-triggered random frames show that LPM is more stable and less distracted by unpredictable inputs than ICM, RND, and other curiosity methods.

**Strengths:**

- The motivation is clear and relevant. It is a known weakness of curiosity-based exploration.
- The method is simple and easy to add to existing RL frameworks.
- Good theoretical grounding: the authors link learning progress to information gain and show the expected-prior term is mathematically justified.
- The experiments are well aligned with the paper’s goal, with clear setups for both state and action noise.
- Consistent improvements over strong baselines on MiniWorld and Atari when noise is introduced

**Weaknesses:**

- The error-predictor network is central to the method, but there are no ablations replacing it with simpler options (e.g., EMA of past errors, or previous model prediction).
- No enough sensitivity analysis for key hyperparameters (queue size, update period, …).
- Evaluation is limited to two noise protocols (CIFAR injection, noisy wall). Other types such as temporal or occlusion noise are not explored.
- Experiments rely only on on-policy methods (A2C/PPO); no test with off-policy RL to check generality.
- Code is not yet released, which hinders reproducibility.
- The paper lacks some references and comparison with prior work. See references as examples.

[References]
Planning to Explore via Self-Supervised World Models — Sekar et al., ICML 2020
BYOL-Explore: Exploration by Bootstrapped Prediction — Guo et al., NeurIPS 2022
Fast and Slow Curiosity for High-Level Exploration in Reinforcement Learning — Nicolas Bougie and Ryutaro Ichise, Applied Intelligence 2021

**Questions:**

1. Add ablations replacing the error-predictor with simpler mechanisms (EMA, previous model) to confirm necessity.
2. Include more types of stochasticity (temporal, occlusion, heavy-tailed) to test robustness.
3. Add off-policy baselines or experiments to check method generality.
4. Release code or add details for reproducibility.
5. Compare to additional noise-aware exploration baselines (e.g., Plan2Explore, Disagreement).

---

> ### Author Response · Authors · 2025-11-20
> **Author rebuttal (I):**
>
> Thank you for your constructive feedback. Below we address the listed concerns:
>
> Q1: Simpler mechanism
>
> This is a critical insight. As we have proved in section 4.2, we use a error prediction model instead of previous model to reduce intrinsic reward variance. And another very important reason that we choose to use an error prediction model is that using only last step model often encounters the situation where state s is not visited since last update, making previous model to be identical with current model when predicting state s, resulting zero intrinsic reward. We believe that using previous model resembles time horizon ensemble methods, which is known to have high computational cost beacuse their intrinsic rewards are often only more useful with more number of historical step models. Error prediction model is a sweet spot in terms of stability and computation efficiency as it can be seen as the ensemble of historical models.
>
> Q2: Temporal or occlusion noise
>
> We chose the two types of stochasticity (state-based and action-based) in our experiments following the existing literature on noise-robust exploration, and they have been proven to be hard enough for existing studies. However, temporal and occlusion noises are indeed possible in the real-world; we look forward to include these type of noises in future studies.
>
> Q3: Off policy baselines.
>
> We follow previous works setting to conduct experiment using PPO and A2C methods. And we also listed detailed LPM’s intrinsic reward behavior to validate our design in Figure 2 and Figure 5. These results directly validate the design of LPM, adding off-policy benchmarks would not change the core conclusions of the paper and is outside the scope of our contribution.
>
> Q4: Code availability
>
> The code to run all experiments will be open-sourced upon publication.
>
> Q5: Noise aware baselines
>
> We would like to clarify that we did include “Self-Supervised Exploration via Disagreement” as our baseline; in table 1, we mistakenly cited the paper that proposed ensemble method instead of citing the correct disagreement paper that use variance of ensembled models to encourage exploration. This has been fixed. And we also included (How to Stay Curious while Avoiding Noisy TVs, 2022) as the latest prediction error based noise aware baseline, and (Episodic Novelty Through Temporal Distance, 2025) as the latest episodic bonus based noise aware baseline. Both methods explicitly discuss noise in their papers.
>
> We thank you again for your support to our work.

---

> > ### Comment · Reviewer_Hppm · 2025-11-21
> >
> > I thank the authors for their thorough comments and for answering my questions.
> > I will maintain my current score as the code is still not available (upon acceptance according to the authors' answer).

---

### Official Review · Reviewer_uP88 · 2025-10-27

**Soundness:** 3
**Presentation:** 2
**Contribution:** 3
**Rating:** 6
**Confidence:** 4

**Summary:**

The paper proposes" Learning Progress Monitoring (LPM)", an exploration method that aims at rewarding an agent for improving its predictions rather than for raw prediction error or novelty. By tracking changes in model error across iterations, LPM wants to focus on learnable transitions and avoids unlearnable noise (e.g., “noisy-TV”).

**Strengths:**

- The contribution tackles an important problem, in particular "(...) strategy is naturally robust to noisy-TV distractions and promotes efficient exploration by directing effort toward the most informative experiences.".
- There is a theoretical connection to information gain to justify the approach and to the necessity of the error model.

**Weaknesses:**

A few elements do not seem to be fully detailed both on the methodology and the experiments.

**Questions:**

*Methodology*
- It is unclear how the approach deals with learning and overfitting in the model. In the context of the noisy TV, does the model fully overfits on the new transition before fitting the error model? How are the hyper-parameters chosen (e.g. training steps, NN architecture, etc.)

*Experiments*
- In Fig 4. the standard deviation of the extrinsic rewards that seems to be reported looks extremely small. Is there a reason? Can you clarify the setup?
-Space Invader, Ms PacMan and UpNDown (noisy) are not the hardest exploration games and are not very sparse in terms of rewards. Could you explain why you choose these games as opposed to e.g. Montezuma's revenge.
- Some visualisations of the intrinsic rewards and the estimated errors on examples (e.g. some specific trajectories of the agent) would help see the inner workings of the algorithm and check the suitability of the experiments.

*Other comments*
- Some typos (e.g. line 144,145, there is no dot between the sentences).

---

> ### Author Response · Authors · 2025-11-20
> **Author rebuttal (I)**
>
> Thank you for your constructive feedback. Below we address the listed concerns:
>
> Q1: Overfitting
>
> In our design, we have theoretically shown our buffer with prediction model can effectively reduce the variance, and the source of variance is exactly because model may overfit to certain steps, causing the intrinsic motivation to fluctuate. We also adopt the usual designs in our experiment to prevent overfitting for all baselines, e.g. batch training and large number of parallel experiments. Our design only have one additional hyperparameter, which is the buffer size for the error predictor, we explicitly set the buffer size to match the step size. As the predictor in LPM is the same as the prediction error based intrinsic motivation methods, we adopt the same neural network setting as AMA method. Details of neural network settings and hyperparameters are shown in the Appendix B. Moreover, following our theoretical analyze, we can show that even overfitting, LPM will not trap the RL agent in noisy states. As both dynamic model and error prediction model overfits immediately, we have dynamic model overfit to $o_t$, error prediction model overfit to $f_{\theta - 1}(o_{t}) = o_{t - 1}$, hence we have intrinsic reward $r_i =  log(MSE(o_{t - 1}) - log(MSE(o_{t})$, as both $o_{t-1}$ and $o_{t}$ are noisy states following $o = s + \epsilon$, we have $r_i = log(\epsilon_{old}^2) - log(\epsilon_{new}^2)$, hence we have $E(r_i) = 0$ matching theorem 4.1 from the paper that $r_i = 0$ when IG = 0.
>
> Q2: Justification of experiment
>
> We use a large number of parallel environments with 128 random seeds. At each training step, we gather 128 transitions from each environment resulting in 128*128=16384 transitions in one training step. As the sample size grow, we should expect standard error to be smaller as a result of the Central Limit Theorem. We build our work upon (How to Stay Curious while Avoiding Noisy TVs), whose work is build upon OpenAI’s (Large-Scale Study of Curiosity-Driven Learning), and we can observe that while their result already have low std with 3 random seeds, our larger number of parallel environments with more random seed settings further reduce the std. Code will be published to reveal all details of our implementation.
>
> We thank you for your comment on the sparsity of our Atari environment. The reason for choosing these environments is that we want to test noise robustness while preserving meaningful gameplay comparing to baselines. And our experiment has shown that our method is capable of motivating gameplay regardless of noise. But we also aware that these games although claimed to be hard games according to reference works (EDT), they are not among the hardest tasks. Hence we have added one new experiment evaluating LPM in an Atari game that commonly considered as the ‘hardest’, Montezuma’s Revenge. We have reported the performance in the paper, and our LPM method surprisingly outperform our baselines by achieving meaningful game play with 20M steps, while baselines require at least 50M steps to achieve any environment rewards. The new Section 5.5 is marked in blue.
>
> Q3. Visualization of reward.
>
> We designed the first experiment aiming to provide a clear view of how LPM intrinsic reward change as more transitions are observed, and has shown that our method is capable of provide consistent intrinsic reward regardless of noise. But we do appreciate your insight that we should further study the behavior in complex tasks. In the newly added Figure 5, we have included the behavior of intrinsic motivation of LPM when exploring Montezuma’s Revenge. The result shows that as the RL agent failed to reach new states at the beginning (~1M to 18M steps), our intrinsic motivation gradually decrease, as no new information is learned during the exploration. And the moment the agent reach a new room (novel states), our intrinsic motivation suddenly rise to encourage the agent to further explore these novel states and eventually capable of score points in the game. Also we would love to show some game play videos upon paper acceptance on the project webpage.
>
> We thank you again for your insightful comments. Your suggestions helped us strengthen our experimental evaluation.

---

### Official Review · Reviewer_CKoM · 2025-11-02

**Soundness:** 3
**Presentation:** 1
**Contribution:** 2
**Rating:** 4
**Confidence:** 4

**Summary:**

This paper introduces Learning Progress Monitoring(LPM), a novel intrinsic reward method to solve the noisy-TV problem where agents get stuck on unlearnable randomness . Instead of rewarding high prediction error, LPM rewards the improvement in prediction accuracy, effectively measuring the model's learning progress . It uses a dual-network to compare the current model's error against the expected error of the previous model. This learning progress reward is theoretically shown to be a monotone indicator of Information Gain (IG), uniquely dropping to zero for both fully learned states and unlearnable states. Empirical results show LPM achieves robust exploration in noisy environments, avoiding the catastrophic failures of other methods while maintaining SOTA performance in deterministic settings.

**Strengths:**

**Novel Reward Mechanism**: This paper proposes an innovative intrinsic reward mechanism based on "learning progress" (improvement in prediction error) rather than the magnitude of the error itself.

**Excellent Noise Robustness**: This draft empirically demonstrated to be unparalleled in handling random uncertainty (noisy TV), preventing the catastrophic performance degradation seen in baselines.

**Strong Theoretical Explanation**: This paper theoretically shows that the LPM reward is a monotonic measure of information gain (IG), mathematically justifying the ignorance of both boring and unlearnable states.

**High Performance in Deterministic Environments**: Unlike specialized methods that can sacrifice general performance, LPM competes with or surpasses state-of-the-art baselines in standard, noise-free environments.

**Fast Adaptation**: In the experiments, the intrinsic reward signal rapidly adapts, converging to zero in both learnable and unlearnable states, which prevents the agent from getting stuck.

**Weaknesses:**

**Not Sufficient Experimental Setup**: The experiments seem to be insufficient as they rely on synthetic noise rather than more realistic, naturally occurring stochasticity. This questions the method's applicability to real-world problems. Therefore, additional explanation and scenarios should be suggested.

**Insufficient and Limited Experiments**: The paper's claims are based on a very small set of environments. Compared with recent researches in RL, the case studies are not satisfactory. More complex and continuous state-based environments could be adopted to extract generalization from experiments. For example, this paper omits critical, hard-exploration benchmarks (like 'Montezuma's Revenge') and lacks testing in procedurally generated or continuous control (robotics) domains.

**Risk of Local Minima in Clean Environments**: In noisy-free settings, the reward mechanism focuses on "learnable" states. This could lead to the agent "getting stuck" by focusing too much on a complex, learnable area it has already found, rather than exploring new, simpler areas (as you pointed out). This intuitive question should be addressed.

**Questions:**

Please, consider concerns in Weakness.

**Details Of Ethics Concerns:**

N.A

---

> ### Author Response · Authors · 2025-11-20
> **Author rebuttal (I)**
>
> Thank you for your constructive feedback and for the positive comments regarding our design and theoretical analysis. Below we address the mentioned weaknesses:
>
> W1: Additional explanation and scenarios for experiment settings:
>
> In this paper, we have classified real world noise into two categories as state based noise and action based noise. Real world noise encountered by RL agent (or presumably a robot) is always a combination of these two type of noise. As exampled in the paper, in a scenario where human robot coexist, unpredictable movement of human can be seen as a state noise, but if the robot keep rotate its camera toward the human, it is also considered as action triggered noise. The goal of the experiment is to comprehensive study how intrinsic motivation methods are affected by these noise. And this experiment setting has provided solid outcome, not only showing that our designed method is robust to both type of noises, but showing existing SOTA, while robust to state noise, does not robust to action based noise, providing a promising insight for future improvements of these existing works.
>
> W2: Insufficient and limited experiment
>
> Thank you for pointing out the issue, we indeed want a more persuasive experiment setting to study the performance of our proposed method across domains. Hence we have added two new experiment sections in the paper: continues control (mountain car) and Montezuma’s Revenge, marked in blue. In continues control experiment, LPM has achieved best exploration coverage in noisy task and have least performance drop because of noise, indicating LPM is capable of filtering out meaningless exploration actions thanking to the monotonic measure of information gain . And surprisingly in Montezuma’s Revenge, LPM also achieved highest exploration efficiency that acquired 400 points (thats all points you can earn in the first room!) within 20M steps, whereas the baseline requires 50M steps. Please refer to newly added Section 5.3 and 5.5 for more details.
>
> W3: Local Minima
>
> This is a interesting question. First, the definition of ‘complex’ and ‘easy’ are generated referring to classic prediction error based (all curiosity methods in table 1) intrinsic motivation problems. As high prediction error is considered as ‘complex’ and will yield high intrinsic motivation, and vice versa for low prediction errors. However things are different in our learning progress method. As LPM is measuring model improvement, the ‘easy’ transition will have a more steep learning curve, hence provide more intrinsic motivation, while the ‘complex’ transition will have a more stable learning curve, hence provide less intrinsic motivation. This can be considered as a continues learning process, encouraging the RL agent to start with ‘easy’ tasks and slowly transit to ‘complex’ tasks.
>
> We thank you again for your insightful comments. Your suggestions helped us strengthen our experimental evaluation, and we have incorporated two additional experiments accordingly.

---

### Author Response · Authors · 2025-12-03
**Summary of Changes and Response to Reviewers (part 1)**

We thank the reviewers (CKoM(4), uP88(6), Hppm(8), vQEu(4)) and the Area Chair for their time and constructive feedback. Below is a summary of our rebuttal period.

1.  **Agreement on Theoretical Soundness.** We first highlight the consensus among all reviewers regarding the novelty and rigor of our theoretical analysis. All reviewers agreed that our derivation of the LPM reward acts as a monotonic indicator of Information Gain that is theoretically robust to stochasticity. This theoretical guarantee forms the solid foundation of our work that our method drops to zero for both fully learned states and unlearnable noise, distinguishing LPM from novelty-based approaches.
2. **Major Experimental Updates: Ensuring Comprehensive Evaluation** The most significant concern raised across reviewers (CKoM, uP88, vQEu) was the need for a broader experimental suite to prove generalization beyond standard noisy environments. In response, we have added three major experiments:
    - **Montezuma’s Revenge (Section 5.5):** To address the need for hard-exploration benchmarks (requested by uP88, vQEu, CKoM), we tested LPM on this notorious task. LPM achieved meaningful exploration and score collection in 20M steps, significantly outperforming baselines of 50M steps, this result is further confirmed by [the CleanRL benchmark](Mikael Henaff), provided by reviewer vQEu. Furthermore, we added Intrinsic motivation
    - **Continuous Control (Section 5.3):** To address the need for continuous state/action validation (requested by CKoM), we added a Continuous MountainCar task with sparse rewards with and without noise. And our LPM method achieved best state coverage and least performance degradation under noise.
    - **MiniGrid-LavaCrossing (Appendix B):** To address the request for classical grid environments (requested by vQEu), we added this MiniGrid-LavaCrossing to Appendix, which is a maze like grid environment with lava grid that will kill the agent immediately. Both ICM, and our LPM method achieved highest performance of ~95% success rate, significantly outperform all baselines.

    With these additions, our experiments cover a diverse set of environments that rigorously test our method across the full spectrum of RL challenges: from varying **visual complexities** (simple grids to high-dimensional pixel images) to continuous vs. discrete action spaces, and sparse vs. rewardless settings. This makes our empirical results robust and comprehensive.

3. **Baseline Adjustments and Explanations:** Several reviewers (vQEu, Hppm) inquired about the inclusion of specific baselines such as Disagreement, NGU, and E3B. We have clarified our selection criteria and corrected a citation error.
    1. We did include the "Disagreement" method (Self-Supervised Exploration via Disagreement) in our original submission (labeled as "Ensemble" in Table 1). We apologize for the initial citation error, which has now been fixed.
    2. We prioritized **EDT (ICLR 2025)** and **EME (NeurIPS 2024)** as our primary episodic bonus as intrinsic motivation baselines. These represent the latest advancements in similarity/count-based methods; notably, the EDT work demonstrated that their method outperforms earlier approaches like NGU, E3B, and DEIR. However, to address specific requests for hard-exploration comparisons, we **added NGU as a baseline** in the new Montezuma's Revenge experiment, as existing baselines failed to report performance on this specific task.

---

> ### Author Response · Authors · 2025-12-03
> **Summary of Changes and Response to Reviewers (part 2)**
>
> Beyond the previous major paper updates, we also list concerns from the reviewers with our responses, please refer to the discussion comments for details:
>
> - **Risk of Local Minima(CKoM):** In our LPM method, "easy" (learnable) transitions yield a steeper learning curve (high intrinsic reward), while "complex" transitions may offer slower progress initially. This naturally guides the agent from easier to harder tasks, forming a continuous learning process.
> - **Overfitting(uP88):** Regarding overfitting, we employ standard mitigation strategies and theoretically demonstrate that our dual-network design is inherently robust: even if overfitting occurs, the dynamic model and error predictor overfit simultaneously to the noise, causing the intrinsic reward to net zero under noise.
> - **Necessity of the Error Predictor(Hppm):** As proven in Section 4.2, the error prediction model is theoretically necessary to reduce variance and guarantee monotonicity with Information Gain. Practically, it serves as a computationally efficient 'sweet spot' as it functions as a lightweight ensemble of previous models, avoiding the high cost of storing multiple previous models.
> - **Validity of Benchmark(vQEu):** We distinct our method from existing RL algorithm studies that use trillions of steps to prove effectiveness. Instead we follow the stochastic study from (How to Stay Curious while Avoiding Aleatoric Uncertainty from ICML 2022) who use 1M step experiment in both Atari and Mario with intrinsic motivation. We specifically study performance degradation under noise across baselines, and all of our experiments has shown LPM is both theoretically and empirically robust to either state based and action based stochasticity.
>
> In summary, our work introduces Learning Progress Monitoring (LPM), a novel intrinsic motivation method that improve RL exploration efficiency while robust to stochasticity by measuring learning progress. We have demonstrated that LPM serves as a theoretically sound proxy for Information Gain and empirically exhibiting the **least relative performance degradation** against noise compared to SOTA baselines. By adding Montezuma’s Revenge, we confirmed that LPM is capable of solving both hard exploration tasks and the Noisy-TV problem, validating it as a generalizable method for intrinsic motivation.

---

### Meta-Review · Area_Chair_Edwz · 2026-01-09

**Summary:**

The paper addresses the Noisy-TV problem in RL by proposing Learning Progress Monitoring (LPM), which rewards model improvement rather than prediction error. While the theoretical contribution - linking LPM to Information Gain - was generally praised for its rigor, there were concerns regarding experimental protocols and thoroughness. Reviewers argued that the 1M step evaluation on Atari is insufficient to draw generalizable conclusions, as it deviates from the 200M step standard and may not capture long-term exploration dynamics (however this was used in prior work as pointed out by reviewers and SAC). Furthermore, there was skepticism regarding the choice of hard exploration games, with reviewers noting that games like Pacman and Space Invaders provide relatively dense rewards. The addition of Montezuma’s Revenge during the rebuttal was noted, but concerns were raised that the PPO+RND baseline was underperforming compared to established benchmarks, potentially inflating the relative gains of the proposed method. An explanation was added during the rebuttal period to address this.

**Reviewer Concerns:**

Addressed:
- There was consensus that the derivation of LPM as a monotonic indicator of Information Gain is novel and mathematically sound.
- The authors successfully added experiments in continuous control (MountainCar) and grid-world (MiniGrid-LavaCrossing) to show broader applicability across different action spaces and visual complexities.

Outstanding Concerns:
- The use of 1M step training seemed too low for solid experimental validation.
- The performance of the RND baseline in the newly added Montezuma's Revenge experiment was deemed problematic but addressed to some extent during the rebuttal period.
- Concerns regarding whether the agent might get stuck in complex but learnable areas rather than continuing to explore simpler unseen regions were raised and not fully settled by the empirical results.

**Reviewer Scores:**

- CKoM: Likely to increase slightly due to the added continuous control and Montezuma's Revenge experiments but the overall protocols were problematic.

- vQEu: This reviewer remained explicitly not convinced by the 1M step Atari results and contested the validity of the Montezuma's Revenge baseline performance.

---

### Decision · Program_Chairs · 2026-01-26

Accept (Poster)